# Serum KL-6 as a Candidate Predictor of Outcome in Patients with SARS-CoV-2 Pneumonia

**DOI:** 10.3390/jcm12216772

**Published:** 2023-10-26

**Authors:** Simone Kattner, Sivagurunathan Sutharsan, Marc Moritz Berger, Andreas Limmer, Lutz-Bernhard Jehn, Frank Herbstreit, Thorsten Brenner, Christian Taube, Francesco Bonella

**Affiliations:** 1Department of Anesthesiology and Intensive Care Medicine, University Hospital Essen, University Duisburg-Essen, 45147 Essen, Germany; 2Center for Interstitial and Rare Lung Disease, Department of Pulmonary Medicine, University Medicine Essen-Ruhrlandklinik, University Duisburg-Essen, 45239 Essen, Germany; 3Department of Pediatric Cardiac Surgery, University Hospital Erlangen, 91054 Erlangen, Germany

**Keywords:** KL-6, COVID-19, SARS-CoV-2 pneumonia, biomarker, disease outcome

## Abstract

Severe acute respiratory syndrome coronavirus type 2 (SARS-CoV-2)-infection is associated with an extremely variable disease course. When interstitial pneumonia (IP) occurs, it can lead to acute respiratory distress syndrome and death. Serum Krebs von den Lungen-6 (KL-6) is an established marker of IP, but its role as a marker of SARS-CoV-2 pneumonia is debated. This bicentric study included 157 patients with SARS-CoV-2 pneumonia. The WHO Ordinal Scale for Clinical Improvement (0–10 points) was used to classify the clinical course. Serum samples were collected at admission, and on days 3 and 7 of hospitalization. KL-6 was measured by using automated chemiluminescence immunoassay. A total of 68 patients developed a severe SARS-CoV-2 pneumonia, 135 of them required oxygen, and 15 died during hospitalization. The patients requiring non-invasive ventilation, invasive ventilation, or extracorporeal membrane oxygenation had significantly higher serum KL-6 levels at admission. The serum KL-6 levels were tendentially higher in patients who died than in those who survived. Logistic regression identified serum KL-6 at a cut-off of 335 U/mL at admission as a significant predictor of severe SARS-CoV-2 pneumonia outcome. Serum KL-6 seems to be a candidate biomarker for the clinical routine to stratify patients with SARS-CoV-2 pneumonia for the risk of a severe disease outcome or death.

## 1. Introduction

Until April 2023, severe acute respiratory syndrome coronavirus 2 (SARS-CoV-2) infected around 763 million people worldwide, causing about 6.8 million deaths [1]. SARS-CoV-2 damages type I and II pneumocytes, causing pneumonia with widely varying degrees of severity [2,3]. An easy-to-obtain, specific biomarker to stratify patients who are at risk of a poor outcome during SARS-CoV-2 pneumonia remains an unmet need.

Several studies have reported patients’ characteristics such as body-mass index; immune-inflammatory parameters, such as procalcitonin (PCT), c-reactive protein (CRP), lactate dehydrogenase (LDH), and interleukin-6 (IL-6); peripheral blood cell counts, such as leucocytes, lymphocytes, neutrophil granulocytes, and thrombocytes; and the ratio between different laboratory parameters, such as the neutrophil–lymphocyte ratio (NLR), as potential predictors for severe SARS-CoV-2 disease [2,4,5,6]. All of these parameters are not disease-specific or lung-specific and often have better prognostic value when considered together.

Krebs von den Lungen-6 (KL-6) is a pneumoprotein of epithelial origin, used to assess disease severity in various pulmonary diseases [7]. In Japan, KL-6 is a validated diagnostic and prognostic marker for interstitial lung diseases (ILD). KL-6 belongs to mucins class 1 (MUC1) and is expressed by type II pneumocytes and respiratory bronchiolar epithelial cells, and along injured epithelial areas. KL-6 serum levels reflect the extent of alveolar damage and can be used to assess ILD severity and predict acute exacerbations [7,8,9]. Serum KL-6 has been investigated in several studies with SARS-CoV-2-infected patients. Although a correlation between serum KL-6 levels and the extent of SARS-CoV-2 pneumonia and related disease prognoses has been observed [10,11,12,13,14], the usefulness of this marker in the clinical management of these patients is still debated [15,16]. In patients with acute respiratory distress syndrome (ARDS), which is similar to acute SARS-CoV-2-induced acute pneumonia, higher KL-6 plasma levels have been found to be associated with increased mortality [17]. In the present study, a larger group investigates whether KL-6 is better than established laboratory parameters at predicting the outcome on the day of hospital admission, whether a cut-off can be identified, whether there is an improvement in prediction with serial KL-6 measurements, and whether a different day improves the predictive capacity of KL-6.

The aim of our retrospective study was to investigate KL-6 levels in the serum of hospitalized patients with SARS-CoV-2 pneumonia, and to evaluate its suitability as a predictor of poor outcome, compared to conventional laboratory parameters.

## 2. Materials and Methods

### 2.1. Study Subjects

We retrospectively studied consecutive patients presenting with SARS-CoV-2 pneumonia at two different institutions belonging to the University of Medicine Essen (University Hospital’s Emergency Department, Intensive Care Unit of the Department of Anesthesiology and Intensive Care Medicine and Pneumology Department at Ruhrlandklinik University Hospital (RLK)). Patients of the Emergency Department and the Intensive Care Unit of the Department of Anesthesiology were grouped under the abbreviation KAI as both departments are at University Hospital. Patients who were hospitalized between April 2020 and August 2021 and gave their written informed consent for data research were included (Appendix A). All patients received a standard coronavirus disease 2019 (COVID-19) therapy according to the available recommendations at that time [18,19,20]. Publications of the RECOVERY Collaborative group and the WHO Living Guideline on Therapeutics for COVID-19 [21] were also included in the treatment strategy. The local ethics committee of the University of Medicine Duisburg-Essen approved the study protocol (20-9216-BO).

### 2.2. Definition and Scoring of Disease Severity

Patient’s outcome was assessed by using the World Health Organization (WHO) Ordinal Scale of Clinical Improvement (score 1–10) [22]. Patients were grouped by the WHO score into two groups: hospitalized with moderate disease (score 4–5 points), hospitalized with severe disease including death (score 6–10 points).

### 2.3. Blood Sampling and Laboratory Assays

Blood was taken on the day of admission (day 0), on day 3, and on day 7 of hospitalization and stored at −80 °C until measurement. Blood was withdrawn within 24 h, mostly before any COVID-19-specific treatment was administered. We cannot exclude that patients had taken other medications at home, such as anti-inflammatory medication or pain relievers. KL-6 was measured in serum samples by using automated chemiluminescence immunoassay (Fujirebio Europe, Gent, Belgium) as previously described [23,24]. Delta KL-6 was calculated as the difference in KL-6 levels between baseline (day 0) and day 7. Other routine laboratory parameters (leucocytes, lymphocytes, neutrophil granulocytes, thrombocytes, procalcitonin (PCT), c-reactive protein (CRP), lactate dehydrogenase (LDH), interleukin-6 (IL-6)) were measured in fresh heparinized blood or plasma. Absolute numbers of leukocytes, lymphocytes, neutrophil granulocytes, and thrombocytes (platelets) were measured by using fluorescence flow cytometry in Sysmex XP300 Automated Hematology Analyzer (Sysmex, Norderstedt, Germany). LDH, PCT, and CRP were analyzed by Atellica Solution Immunoassay & Clinical Chemistry Analyzers (Siemens Healthineers International AG, Zurich, Switzerland). IL-6 was determined with a sequential solid-phase chemiluminescence immunoassay by IMMULITE 2000 XPI Immunoassay Analyzer (Siemens Healthineers International AG, Zurich, Switzerland). The neutrophil–lymphocyte ratio (NLR) was calculated.

### 2.4. Statistical Analysis

Normally distributed variables are presented as mean ± standard error of the mean (SEM) or as median with interquartile range (IQR) if not normally distributed. Categorial variables are presented as percentages of the total in the category and as numerical. The 3-fold interquartile range was used to determine outliers from the severe and moderate disease groups. Non-normal distribution was observed for all variables except body-mass index (BMI), LDH, CRP, and neutrophil count in Kolmogorov–Smirnoff test. Comparison between groups was made by using *t*-test or ANOVA for continuous variables and Chi-square or Fisher’s exact test for categorial variables if normally distributed, and the Mann–Whitney U test was used to compare non-normally distributed data. The Spearman correlation was calculated to examine the variables at hospital admission for a correlation with a possible later admission to the intensive care unit. A *p*-value < 0.05 was assumed as significant. Receiving operating curves (ROC) analysis was used to test the role of each variable to predict a severe disease outcome. The cut-off values were determined with the minimum requirement of a sensitivity of 60 and a 1-specificity of less than 40. Binary logistic regression was used to investigate predictors of disease outcome.

IBM SPSS Statistics was used for analysis (IBM Corp. Released 2020. IBM SPSS Statistics for Windows, Version 27.0. Armonk, NY, USA: IBM Corp).

## 3. Results

### 3.1. Characteristics of Study Subjects

The demographics and laboratory parameters of the subjects at admission are shown in Table 1. In total, 157 patients were included in the study; of them, 96 (61%) were males. The patients were, on average, 61 [49–73.5] years old. Apart from a higher BMI in the patients admitted to the Pneumology Department at Ruhrlandklinik University Hospital (RLK), there were no relevant differences in demographics between the two recruiting institutions.

Table 2 shows the demographic and laboratory characteristics of the studied patients according to the disease severity. A total of 89 patients had a moderate disease (WHO score 4–5) with no need for intensive care treatment and 68 patients were admitted to an intensive care unit (ICU) with a severe disease (WHO Score 6–9). Twenty patients (13%) were intubated, and more frequently in the emergency department. In total, 15 patients died, due to ARDS and sepsis with multiple-organ failure (MOF).

There was a male prevalence in the moderate disease as well as in the severe disease group (53% and 72%, respectively). The ages were similar in the two groups with averages of 61 [48–73] years in the moderate disease group and 62.5 [50–74] years in the severe disease group (*p* = 0.403). The average BMI in the severe group was significantly higher (25 ± 1 kg/m^2^ vs. 28 ± 1 kg/m^2^, *p* = 0.06). Serum biomarkers such as PCT (*p* < 0.001), CPR (*p* = 0.012), and IL-6 (*p* = 0.013) showed higher average values in the severe disease group. Also, the LDH, NLR, and leukocyte count showed a slight tendency to a higher average value in the severe disease group.

### 3.2. Serum KL-6 Levels

At baseline, the average serum KL-6 level was 407 [260.1–627] U/mL with no substantial difference between the two recruiting institutions. Intubated patients had a higher KL-6 value of 580.5 [IQR 351–922] U/mL than the non-intubated patients, with a KL-6 value of 401 [IQR 258.5–607] U/mL (*p* = 0.02) (Figure 1). The patients who died tended to have a higher KL-6 value on admission compared to the survivors (608 [IQR 348–935] U/mL vs. 404 [IQR 259–609] U/mL, respectively, *p* = 0.06).

The mean KL-6 level at baseline was 364 [IQR 245–511.5] U/mL in the moderate disease outcome group (WHO score 4–5) and 542.5 [IQR 350–838.3] U/mL in the severe group (WHO Score 6–9) (*p* = 0.001) (Figure 2). The serum KL-6 levels of patients with serial measurements on day 3 and on day 7 are shown in Figure 3. On day 3, elevated median KL-6 values were seen in both disease outcome groups (Table 2). On day 7, the KL-6 value tended to be higher in the severe group (457.5 [IQR 328–657.5] U/mL) vs. the moderate disease outcome group 631.5 [IQR 460.8-1018] U/mL, respectively, *p* = 0.013). The difference in KL-6 levels between baseline (day 0) and day 7 (Delta KL-6) was higher in patients with severe disease than in those with a moderate disease (67.5 [IQR 24.3–150.8] U/mL vs. 631.5 [IQR 460.8–1018] U/mL, respectively, *p* = 0.018). Inflammatory parameters, such as PCT, CRP, and IL-6, were also higher in the severe disease group than in the moderate disease group (Table 2).

The baseline KL-6 levels exhibited a positive correlation with the LDH levels (r = 0.244, *p* = 0.004), while the IL-6, PCT, CRP, NLR, and P/F levels did not show any significant correlation (*p* > 0.05) (Table 3).

### 3.3. Predictive Value of Baseline Serum KL-6 and Other Biomarkers for SARS-CoV-2 Pneumonia Severity

An ROC analysis was performed for KL-6 and the LDH, NLR, PCT, CRP, and thrombocytes (Table 4 and Figure 4) in order to identify cut-off values associated with disease severity. The identified cut-off values are shown in Table 4. Except for the LDH, NLR, and thrombocyte count, all other biomarkers showed from a moderate to a good ability (AUC > 0.70) to predict severe SARS-CoV-2 pneumonia outcome at admission. For serum KL-6, the cut-off with the best sensitivity and specificity was 335 U/mL (0.8 and 0.6, respectively, AUC 0.70). Mortality could not be predicted by baseline KL-6 or any other laboratory parameter in this study. A KL-6 value of >335 U/mL on the day of hospital admission correlated positively with a later admission to the ICU (Table 5).

### 3.4. Logistic Regression for Predictors of Disease Severity

Binary logistic regression was performed to investigate predictors of a severe SARS-CoV-2 pneumonia outcome. In the first regression, we included the selected biomarkers as continuous variables together with demographic and clinical variables (Table 6). After four backward conditional steps, only the BMI, NLR, PCT, and LDH were significant predictors of the disease outcome after adjustment for demographics (Table 6).

In a following regression analysis, we included cut-off values for each biomarker, obtained by ROC analysis (Table 4). The results of the logistic regression for cut-off biomarkers are shown in Table 7. After three backward conditional steps, the BMI, a KL-6 levels > 335 U/mL, CRP levels > 5.2 mg/dL, PCT levels > 0.075 ng/mL, and NLR levels > 3.5 were still significant predictors of disease outcome after adjustment for several covariates (Table 6). Compared to other biomarkers included in the analysis, serum KL-6 levels higher than 335 U/mL were associated with the highest OR for severe disease outcome or death (OR 4.642, 96% CI 1.457–14.786).

## 4. Discussion

In the present study, we show that serum KL-6 levels at hospital admission are higher in patients with SARS-CoV-2 pneumonia and severe outcomes. Additionally, we identified a cut-off of serum KL-6 with the potential to predict a severe SARS-CoV-2 pneumonia outcome compared to other routine biomarkers.

Although several studies have been recently conducted, the role of serum KL-6 as a biomarker of SARS-CoV-2 pneumonia and its outcome remains poorly understood in the European population [10,12,13,14,15,16]. This bicentric study has tried to investigate serum KL-6 as a biomarker of SARS-CoV-2 pneumonia severity and outcome, comparing it with other routine laboratory markers. The mean levels of serum KL-6 in our cohort (407 [260.1–627] U/mL) were comparable with those described in other studies [10,13,14]. Slight differences in serum KL-6 levels can be explained by different methods used to measure KL-6 (we used a fully automated CLIA system) or the genetic background of the studied populations. We did not find any significant correlation between the serum KL-6 levels and demographics or other routine laboratory markers (Table 3) in our cohort except for a weak correlation (r < 0.5) with serum LDH. LDH is a non-specific marker for cell decay and is used in the routine management of patients with respiratory diseases, especially in ILD, for disease severity assessment. Due to the weakness of the correlation, we decided not to further explore it in this study. The lack of significant correlations between KL-6 levels and demographics has already been pointed out in other studies [25,26,27]. This can be seen as an advantage, that KL-6 levels do not depend on age, gender, BMI, or smoking status, thus reducing the possible influence of covariates on the results of the present study. Similarly to other recently published studies, we also found that LDH, CRP, NLR, P/F, and PCT correlated with SARS-CoV-2 pneumonia severity and outcome [4,6]. These markers reflect different stages of lung injury and inflammation, and their serum levels could also depend on the kinetics of each single biomarker. For serum KL-6, we know that the marker is lung epithelial-specific, accumulates in the alveolar room, and spills over into the blood when the permeability of the alveolar arterial barrier is augmented [7].

In terms of repeated measurements of KL-6 during hospitalization, we found that the serum KL-6 levels on day 3 after admission were equally high in both the moderate and severe groups. KL-6 is essentially produced by regenerating pneumocyte type II and gets into the blood by increased permeability following disintegration of the alveolar–vessel barrier after injury [7]. The kinetics of elevation of KL-6 in serum over time has not been sufficient explored, but we hypothesize a similar pathway as for the acute exacerbation of idiopathic pulmonary fibrosis [9]. We cannot exclude that treatment initiated between day 1 and 3 could have significantly affected the elevation of the KL-6 levels in the serum. Due to the low number of serial KL-6 measurements, we cannot provide an unequivocal interpretation of these findings. Only in patients with a severe outcome or death did KL-6 levels remain high until day 7. Previous studies report contrasting data on short-term variations in KL-6 [14,28,29,30]. The study by d’Alessandro et al. [29] had longer sampling intervals and cannot be compared to our study. Other studies showed a further decrease in serum KL-6 levels in patients with mild or moderate disease. Regarding long-term variations in serum KL-6 levels, Awano et al. and Xue et al. [14,30] observed persistently high serum KL-6 levels even after pneumonia resolution, particularly in patients who developed fibrotic interstitial lung disease [28,29]. A long term follow-up study to confirm these relevant findings in our cohort is warranted.

In terms of prognosis, the logistic regression identified serum KL-6 at a cut-off 335 U/mL, but not as a continuous variable, as a predictor of SARS-CoV-2 pneumonia outcome. As of now, different cut-off values for serum KL-6 have been described [10,14,31]. In line with our results, almost all of them were above 300 U/mL. D’ Alessandro et al. [10], Awano et al. [14], and Chen et al. [31] identified slightly higher cut-offs than in our study. We observed an 80% sensitivity and a 57% specificity for serum KL-6 to predict severe SARS-CoV-2 pneumonia outcome, which is comparable to the other studies [10,14,31]. With an OR of 4.642, serum KL-6 was the strongest predictor of a severe disease among the considered biomarkers. Several studies with smaller sample sizes are in line with our findings [10,14,29], but some others did not confirm KL-6 as an independent predictor of poor outcome [15,16,32,33]. The inconsistency of the results across studies could depend on the different classifications used for defining SARS-CoV-2 pneumonia severity. In some studies, for example, the moderate or the severe group are defined very broadly, so that the considered groups considerably overlap or the severe group does not include any intensive care patients [16,32,33]. In other studies, like that by Scotto et al. [13], a very high OR of 17 for the serum KL-6 cut-off of >1000 U/mL was observed, but the sample size was only 34 patients. It is worth mentioning that, in our study, CRP, PCT, LDH, and NLR, in line with other investigations, were also good predictors for SARS-CoV-2 pneumonia outcome [4,6,12]. The advantage of KL-6 compared to other routine biomarkers is that concomitant bacterial coinfections, sepsis, or steroid treatment do not significantly influence its serum levels, hence potentially avoiding misinterpretations [7].

Although the strength of this study is that patients from two different institutions were included, potentially minimizing selection biases, there are several limitations to mention. First, the small sample size did not allow for performing subgroup analyses. Second, we did not employ power analysis, thus we do not know if the statistical power is sufficient. Third, we did not systematically review high-resolution CT scans, so we were not able to adjust the predictors for pneumonia extent. Fourth, this was a retrospective exploratory analysis. Finally, the serial measurements of KL-6 during hospitalization were performed only in a few patients, limiting the value of our longitudinal analysis. A further prospective to implement biomarker studies is AI algorithms, such as artificial neuronal networks, for tracking dynamic changes. We learned from several studies that automated machine learning algorithms improve the finding of clinical diagnoses by being objective and more efficient. This can facilitate the daily clinical routine and can contribute to finding reliable biomarkers. AI algorithms have the potential to significantly improve research methodologies and, as a result, patient care and the quality of healthcare delivery. Consideration should be given to this in future biomarker studies.

## 5. Conclusions

In conclusion, serum KL-6 seems to be a candidate biomarker to stratify patients with SARS-CoV-2 pneumonia for the risk of severe disease outcome or death. Further validation of these results is needed.

## 6. Patents

All subjects gave their informed consent for inclusion before they participated in the study. The study was conducted in accordance with the Declaration of Helsinki, and the protocol was approved by the Ethics Committee of University of Medicine Duisburg-Essen (Project identification code 20-9216-BO).

## Figures and Tables

**Figure 1 jcm-12-06772-f001:**
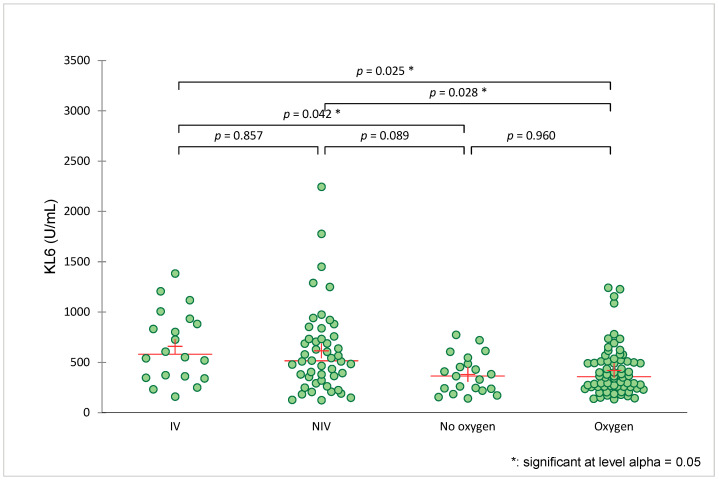
Serum KL-6 levels according to the type of oxygen application or ventilation. Legend: green dot: patient; red line: median; red cross: average; IV: invasive ventilation; NIV: non-invasive ventilation; no oxygen: room air; oxygen: administered by mask or nasal prongs.

**Figure 2 jcm-12-06772-f002:**
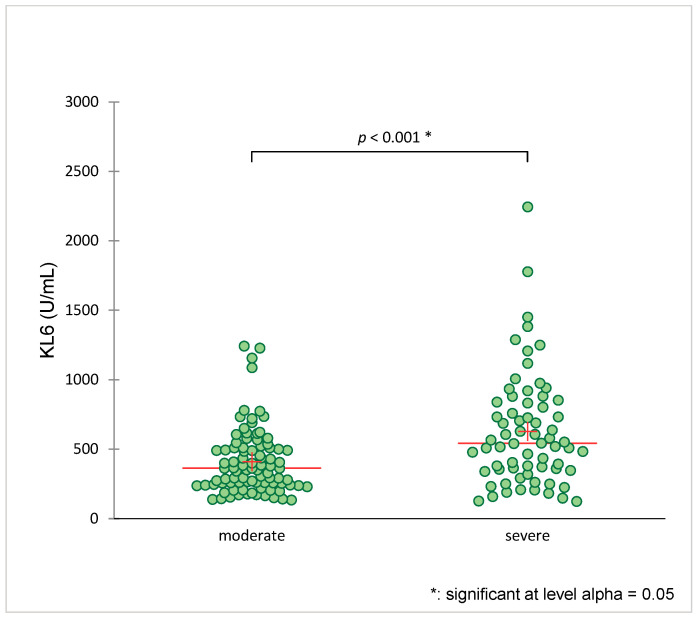
Serum KL-6 levels according to SARS-CoV-2 pneumonia disease group. Legend: green dot: patient; red line: median; red cross: average; Patients were stratified by the World Health Organization (WHO) Ordinal Scale of Clinical Improvement (score 1–10) into hospitalized with moderate disease (score 4–5 points) and hospitalized with severe disease including death (score 6–10 points).

**Figure 3 jcm-12-06772-f003:**
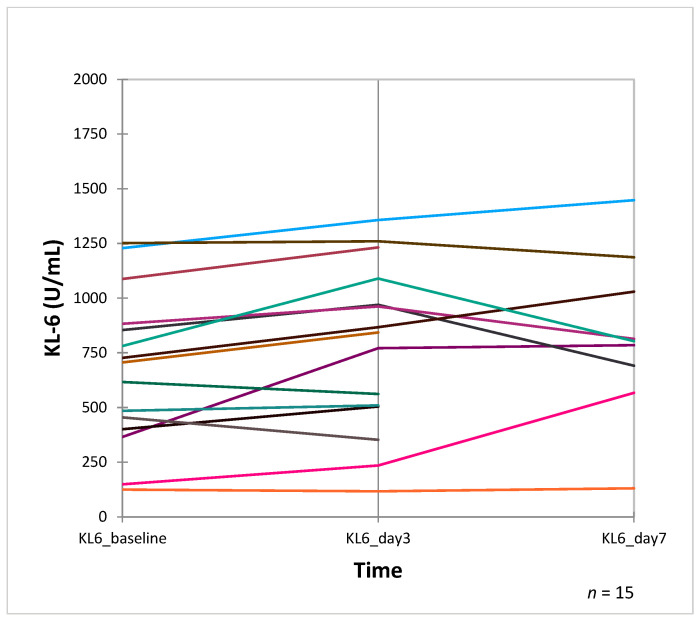
Serum KL-6 levels in patients with serial measurement. Legend: Blood was taken on the day of admission (baseline), on day 3 and on day 7 after admission. Each colored line represents single patients and their serial serum KL-6 measurements.

**Figure 4 jcm-12-06772-f004:**
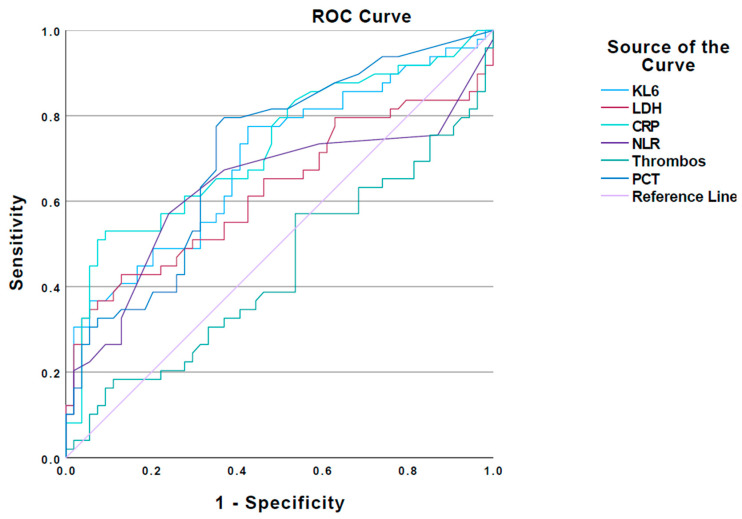
ROC analysis for predicting SARS-CoV-2 pneumonia severe outcome. Receiving operating curve (ROC) analysis of the investigated serum biomarkers for prediction of SARS-CoV-2 pneumonia severe outcome (WHO = 6–10). AUC, best cut-offs and significance are shown in Table 4.

**Table 1 jcm-12-06772-t001:** Demographics, outcomes, and characteristics of all enrolled subjects, stratified according to the enrolling institution.

Variables	All (*n* = 157)	RLK (*n* = 66)	KAI (*n* = 91)	*p*-Value
Age, yrs.	61 [49–73.5]	59.5 [48.5–70.3]	62 [49–76]	0.287
Gender, M/F	96/61	40/26	56/35	0.906
BMI kg/m^2^	26 ± 1	30 ± 1	25 ± 1	0.01 *
SARS-CoV-2 severity/outcome				
WHO score (mean)	6	5	6	0.38 ^+^
Oxygen (yes/no)	135/22 (86%/14%)	54/12 (82%/18%)	81/10 (89%/11%)	0.25 ^+^
ICU admission (yes/no)	68/89 (43%/57%)	26/40 (39%/61%)	42/49 (46%/54%)	0.4 ^+^
IV (yes/no)	20/137 (13%/87%)	4/62 (6%/94%)	16/75 (18%/82%)	0.05 ^+^
Death (yes/no)	15/142 (10%/90%)	3/63 (5%/95%)	12/79 (13%/87%)	0.07 ^+^
Blood cell count				
Leucocyte, ×10^3^/µL	7 [5.2–9.95]	7.2 [5.6–10.2]	6.8 [4.8–9.6]	0.252
Lymphocyte, ×10^3^/µL	1.2 [0.7–1.7]	1.5 [1.2–2]	1 [0.6–1.5]	0.002
Neutrophil, ×10^3^/µL	5 ± 0	5 ± 0	5 ± 0	0.34 *
Thrombocyte, ×10^3^/µL	226 [169.3–326]	321 [238.8–421]	186 [126.5–240.5]	0.000
Serum biomarkers				
KL-6, U/mL at baseline	407 [260.1–627]	501.5 [291.3–742.5]	381 [250–544]	0.038
KL-6, U/mL day 3	843 [505–1090] (n = 15)	855 [467–1125] (n = 14)	n.a. (n = 1)	n.a.
KL-6, U/mL day 7	553.5 [370.8–789.5] (n = 50)	653 [429.8–868] (n = 14)	522.5 [366.3–773.5] (n = 36)	0.336
Delta KL-6, U/mL baseline to day 7	123 [29.5–270.3] (n = 50)	57 [20.3–177] (n = 14)	147 [30–348] (n = 36)	0.120
LDH, U/L	337 ± 15 (n = 138)	312 ± 13 (n = 52)	352 ± 22 (n = 86)	0.13 *
IL-6, pg/mL	22.4 [8.1–69.5] (n = 87)	7.8 [5.1–31.4] (n = 27)	33.4 [12.4–86.3] (n = 60)	0.000
PCT, ng/mL	0.11 [0.03–0.4] (n = 115)	.04 [0–4.8] (n = 31)	0.13 [0.1–0.9] (n = 84)	0.000
CRP, mg/dL	46 ± 11 (n = 143)	5 ± 1 (n = 56)	71 ± 16 (n = 87)	0.01 *
NLR	3 [2–6] (n = 127)	3 [2–5] (n = 42)	6 [3–8] (n = 85)	0.889
P/F	112 [64–167.3] (*n* = 44)	(*n* = 0)	112 [64–167.3] (*n* = 44)	n.a.

Normally distributed data presented as mean ± SEM and non-normally distributed data presented as median [25 quartile–75 quartile]. Deviating numbers due to missing values are indicated in brackets (*n*). Laboratory analyses were collected on hospital admission. Delta KL-6 is the difference in KL-6 levels between baseline (day 0) and day 7. Abbreviations: BMI: body mass index; KAI: Emergency Department at the University Hospital Essen and the intensive care unit of the Department of Anesthesiology and Intensive Care Medicine; ICU: intensive care unit; IV: invasive ventilation; n.a.: not applicable; Il-6: interleukin 6; LDH: lactate dehydrogenase; NLR: neutrophil-lymphocyte-ratio; PCT: procalcitonin; P/F: arterial partial pressure of oxygen divided by the inspired oxygen concentration; RLK: Pneumology Department at Ruhrlandklinik University Hospital; * *p*-values obtained by *t*-test; all other asymptotic signification by Mann–Whitney U test unless otherwise indicated; ^+^: chi^2^ test.

**Table 2 jcm-12-06772-t002:** Demographics and characteristics of the studied subjects stratified by disease severity according to WHO Ordinal Scale of Clinical Improvement (score 4–10) [22].

Variables	Moderate Disease WHO Score 4–5 (*n* = 89)	Severe Disease WHO Score 6–10 (*n* = 68)	*p*-Value
Age, yrs	61 [48–73]	62.5 [50–74]	0.403
Gender, M/F	47/42	49/19	0.02
BMI kg/m^2^	25 ± 1	28 ± 1	0.006 *
SARS-CoV-2 severity/outcome			
Oxygen (yes/no)	67/22 (75%/25%)	68/0 (100%/0%)	<0.001 ^+^
ICU admission (yes/no)	0/89 (0%/100%)	68/0 (100%/0%)	n.a.
IV (yes/no)	0/89 (0%/100%)	20/48 (30%/70%)	<0.001 ^+^
Death (yes/no)	0/89 (0%/100%)	15/53 (23%/77%)	<0.001 ^+^
Blood cell count			
Leucocyte, ×10^3^/µL	6.3 [4.5–8.8]	8 [6–10.6]	0.014
Lymphocyte, ×10^3^/µL	1.3 [0.8–1.8]	0.9 [0.6–1.5]	0.010
Neutrophil, ×10^3^/µL	4 ± 0	5 ± 0	0.019 *
Thrombocyte, ×10^3^/µL	239 [170.8–327]	214.5 [136–313.3]	0.196
Serum biomarkers			
KL-6, U/mL at baseline	364 [245–511.5]	542.5 [350–838.3]	<0.001
KL-6, U/mL day 3	826 [467–1263.3] (*n* = 6)	843 [372.5–966] (*n* = 9)	0.556
KL-6, U/mL day 7	457.5 [328–657.5] (*n* = 22)	631.5 [460.8–1018] (*n* = 28)	0.013
Delta KL-6, U/mL baseline to day 7	67.5 [24.3–150.8] (*n* = 22)	179 [47.8–434] (*n* = 28)	0.018
LDH, U/L	309 ± 12 (*n* = 81)	377 ± 30 (*n* = 57)	0.04 *
IL-6, pg/mL	19.4 [8.2–38.8] (*n* = 48)	49.3 [8.1–97] (*n* = 39)	0.013
PCT, ng/mL	0.05 [0–0.25] (*n* = 60)	0.2 [0.1–2] (*n* = 55)	<0.001
CRP, mg/dL	21 ± 9 (n = 79)	77 ± 20 (n = 64)	0.012 *
NLR	3 [2–4] (n = 72)	5 [1–8] (n = 55)	0.016
P/F	146 [43.8–191.3] (*n* = 8)	109.5 [64–155.5] (*n* = 36)	0.447

Normally distributed data presented as mean ± SEM and non-normally distributed data presented as median [25 quartile–75 quartile]. Deviating numbers due to missing values are indicated in brackets (*n*). Laboratory analyses were collected on hospital admission. Delta KL-6 is the difference in KL-6 levels between baseline (day 0) and day 7. Abbreviations: BMI: body mass index; ICU: intensive care unit; IV: invasive ventilation; n.a.: not applicable; Il-6: interleukin 6; LDH: lactate dehydrogenase; NLR: neutrophil-lymphocyte-ratio; PCT: procalcitonin; P/F: arterial partial pressure of oxygen divided by the inspired oxygen concentration; * *p*-values obtained by *t*-test; all other asymptotic signification by Mann–Whitney U test unless otherwise indicated; ^+^: chi^2^ test.

**Table 3 jcm-12-06772-t003:** Correlations of serum KL-6 with demographics and clinical characteristics at hospital admission.

	*n*	Correlation Coefficient (r)	*p*
Age years	157	−0.052	0.52
BMI kg/m^2^	124	0.164	0.068
Blood cell count			
Leucocyte, ×10^3^/µL	155	0.087	0.281
Lymphocyte, ×10^3^/µL	131	0.151	0.086
Neutrophil, ×10^3^/µL	128	0.199	0.024
Thrombocyte, ×10^3^/µL	152	0.134	0.099
Serum biomarkers			
LDH, U/L	138	0.244	0.004
IL-6, pg/mL	87	0.114	0.114
PCT, ng/mL	115	0.039	0.680
CRP, mg/dL	143	0.036	0.671
NLR	127	0.069	0.441
P/F	44	0.272	0.074

Correlation coefficients were calculated by Pearson coefficient. Abbreviations: BMI: body mass index; Il-6: interleukin 6; LDH: lactate dehydrogenase; NLR: neutrophil-lymphocyte-ratio; PCT: procalcitonin; P/F: arterial partial pressure of oxygen divided by the inspired oxygen concentration.

**Table 4 jcm-12-06772-t004:** ROC analysis obtained cut-offs for each serum biomarker, with the best sensitivity and specificity for predicting SARS-CoV-2 pneumonia severe outcome (WHO = 6–10).

Variables	Cut-Off	AUC	S.E.	Se	Sp	*p* Value
KL-6, U/mL baseline	335	0.70	0.052	0.80	0.57	0.001
LDH, U/L	308	0.63	0.057	0.65	0.54	0.028
NLR, ratio	3.5	0.64	0.058	0.67	0.63	0.018
PCT, ng/mL	0.075	0.71	0.051	0.80	0.59	0.000
CRP, mg/dL	5.2	0.73	0.051	0.80	0.52	0.000
Thrombocyte, ×10^3^/µL	n.a.	0.44	0.058	n.a.	n.a.	0.298

Abbreviations: AUC: area under the curve; S.E.: standard error; Se: sensitivity; Sp: specificity.

**Table 5 jcm-12-06772-t005:** Demographics and characteristics of the studied subjects stratified by KL-6 cut-off 335 U/mL.

Variables	KL-6 < 335 U/mL (*n* = 56)	KL-6 ≥ 335 U/mL (*n* = 101)	*p*-Value
Age, yrs	64 [50.3–76.8]	59 [48–72]	0.248
Gender, M/F	30/26	66/35	0.148
BMI kg/m^2^	25 ± 1	27 ± 1	0.124 *
SARS-CoV-2 severity/outcome			
WHO score (mean)	5	6	0.002 ^+^
Oxygen (yes/no)	46/10 (82%/18%)	89/12 (88%/12%)	0.301 ^+^
ICU admission (yes/no)	15/41 (27%/73%)	53/48 (52%/48%)	0.002 ^+^
IV (yes/no)	3/53 (5%/95%)	17/84 (17%/83%)	0.046 ^+^
Death (yes/no)	2/54 (4%/96%)	13/88 (13%/87%)	0.087 ^+^
Blood cell count			
Leucocyte, ×10^3^/µL	6.8 [4.8–9.3]	7.3 [5.3–10.2]	0.402
Lymphocyte, ×10^3^/µL	1.2 [0.8–1.6]	1.2 [0.6–1.9]	0.888
Neutrophil, ×10^3^/µL	4 ± 0	5 ± 0	0.348 *
Thrombocyte, ×10^3^/µL	199.5 [166–303.2]	244 [170–353.5]	0.137
Serum biomarkers			
KL-6, U/mL at baseline	235.5 [180.3–262.8]	553 [432.5–767]	<0.001
KL-6, U/mL day 3	n.a. (n = 2)	867 [536–1161] (n = 13)	0.027
KL-6, U/mL day 7	279 [229.3–477.3] (n = 16)	672.5 [508.5–954.3] (n = 34)	<0.001
Delta KL-6, U/mL baseline to day 7	73.5 [17.5–235.5] (n = 16)	161 [34–281.3] (n = 34)	0.14
LDH, U/L	315 ± 16 (n = 52)	355 ± 21 (n = 86)	0.137 *
IL-6, pg/mL	27.9 [8–64.2] (n = 36)	21.7 [8.3–84.1] (n = 51)	0.695
PCT, ng/mL	0.01 [0.02–0.3] (n = 47)	0.13 [0.04–0.5] (n = 68)	0.042
CRP, mg/dL	23 ± 11 (n = 52)	60 ± 15 (n = 91)	0.053 *
NLR	3 [2–5] (n = 50)	3 [2–6] (n = 77)	0.613
P/F	126.5 [90.8–173.5] (*n* = 12)	108.3 [62–161.3] (*n* = 32)	0.392

Normally distributed data presented as mean ± SEM and non-normally distributed data presented as median [25 quartile–75 quartile]. Deviating numbers due to missing values are indicated in brackets (*n*). Laboratory analyses were collected on hospital admission. Delta KL-6 is the difference in KL-6 levels between baseline (day 0) and day 7. Abbreviations: BMI: body mass index; ICU: intensive care unit; IV: invasive ventilation; n.a.: not applicable; Il-6: interleukin 6; LDH: lactate dehydrogenase; NLR: neutrophil-lymphocyte-ratio; PCT: procalcitonin; P/F: arterial partial pressure of oxygen divided by the inspired oxygen concentration; * *p*-values obtained by *t*-test; all other asymptotic signification by Man–Whitney U test unless otherwise indicated; ^+^: chi^2^ test.

**Table 6 jcm-12-06772-t006:** Logistic regression analysis for predictors (continuous variables) of a severe SARS-CoV-2 pneumonia.

Variables	B	S.E.	Wald	*p* Value	Odds Ratio	95% C.I. for OR	
						Lower	Upper
Univariate							
Age, yrs.	0.009	0.018	0.248	0.618	1.009	0.974	1.046
Gender (Male)	−0.836	0.622	1.805	0.179	0.433	0.128	1.467
BMI, kg/m^2^	0.093	0.060	2.456	0.117	1.098	0.977	1.234
KL-6, U/mL, baseline	0.001	0.001	1.959	0.162	1.001	0.999	1.003
LDH, U/L	0.007	0.003	6.634	0.010	1.007	1.002	1.013
NLR	0.175	0.070	6.331	0.012	1.192	1.040	1.366
PCT, ng/mL	0.019	0.016	1.367	0.242	1.019	0.987	1.052
CRP, mg/dL	0.001	0.008	0.007	0.931	1.001	0.985	1.017
Multivariate *							
BMI, kg/m^2^	0.130	0.052	6.226	0.013	1.139	1.028	1.261
NLR	0.172	0.066	6.787	0.009	1.187	1.043	1.351
PCT, ng/mL	0.021	0.006	12.944	0.001	1.021	1.009	1.032
LDH, U/L	0.009	0.003	9.770	0.002	1.009	1.003	1.014

* Calculated by using backward stepwise conditional method (five steps).

**Table 7 jcm-12-06772-t007:** Logistic regression analysis for predictors of a severe SARS-CoV-2 pneumonia evaluated by cut-offs.

Variables	B	S.E.	Wald	*p* Value	Odds Ratio	95% C.I. for OR	
						Lower	Upper
Univariate							
Age, yrs.	0.020	0.018	1.256	0.262	1.021	0.985	1.058
Gender (Male)	−0.646	0.595	1.188	0.276	0.523	0.163	1.679
BMI, kg/m^2^	0.143	0.058	5.993	0.014	1.153	1.029	1.293
KL-6 cut-off 335 U/mL, baseline	1.569	0.620	6.412	0.011	4.800	1.425	16.166
LDH cut-off 308 U/L	0.352	0.715	0.243	0.622	1.422	0.350	5.777
NLR cut-off 3.5	0.692	0.635	1.188	0.276	1.997	0.576	6.929
PCT cut-off 0.075 ng/mL	1.399	0.634	4.867	0.027	4.050	1.169	14.033
CRP cut-off 5.2 mg/dL	1.276	0.718	3.162	0.075	3.583	0.878	14.629
Multivariate *							
BMI, kg/m^2^	0.135	0.051	6.899	0.009	1.144	1.035	1.266
KL-6 cut-off 335 U/mL, baseline	1.535	0.591	6.787	0.009	4.642	1.457	14.786
NLR cut-off 3.5	0.953	0.578	2.719	0.099	2.594	0.835	8.058
PCT cut-off 0.075 ng/mL	1.374	0.611	5.057	0.025	3.952	1.193	13.092
CRP cut-off 5.2 mg/dL	1.398	0.661	4.473	0.034	4.047	1.108	14.784

* Calculated by using backward stepwise conditional method (four steps).

## Data Availability

The datasets analysed during the current study are available from the corresponding author on reasonable request.

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
