# Peer review of "Serum KL-6 as a Candidate Predictor of Outcome in Patients with SARS-CoV-2 Pneumonia"

_jcm, 2023, doi:10.3390/jcm12216772_

Round 1

Reviewer 1 Report

Comments and Suggestions for Authors

Review of the manuscript “Serum KL-6 as a candidate predictor of outcome in patients with SARS-CoV-2 pneumonia”

The objective of the study was to to examine the KL-6 levels in serum of hospitalized patients with SARS-CoV-2 pneumonia, and to evaluate its suitability as a predictor of poor outcome, compared to conventional laboratory parameters. Overall, the manuscript is well-structured and clearly written, with a good command of English and clear representation of the aim of the paper. I could recommend its publication in the Journal of Clinical Medicine after considering the following issues:

Abstract: The abstract summarizes the major aspects of the entire paper, including the overall purpose of the study and the research problem, the basic design of the study, major findings and a brief conclusion.

Introduction: The introduction provides a clear and concise overview of the research problem, establishing its relevance in the field. The opening paragraph effectively captures the reader's attention by highlighting the significance of the topic. However:

§  Lines 33-34: “An easy-to-obtain biomarker to stratify patients at risk of poor outcome during SARS-CoV-2 pneumonia remains an unmet need.” Which laboratory parameters have been utilized up to this point, particularly those that are part of routine clinical laboratory diagnostics? E.g. Refer to PMID: 35976368 and 36371787. Highlighting the importance of this information is crucial because understanding which laboratory parameters have been used and why they are relevant provides valuable context for the study and helps establish the basis for comparing the effectiveness of KL-6 in clinical assessment. This knowledge aids in assessing the significance and potential clinical impact of the research findings.

Methods: The study methods are valid and reliable. The process of subject selection is clear. Variables are defined and measured appropriately. The authors employed appropriate statistical methods for the treatment of the data obtained. However:

§  Did author employe the power analysis?

§  Which method was used to determine whether sample data has been drawn from a normally distributed population?

§  Please provide explicit specifications for both the criteria that should be considered for inclusion and those that should be excluded.

§  Lines 81-83: “Other routine laboratory parameters (leucocytes, lymphocytes, neutrophil granulocytes, thrombocytes, procalcitonin (PCT), c-reactive protein (CRP), lactate dehydrogenase (LDH), interleukin-6 (IL-6) were measured in fresh heparinized blood or plasma.” Please provide information on the methodology and the specific analyzer used to determine these parameters.

§  Line 78: “Blood was taken on the day of admission.” Was there any specific treatment given to the patients upon admission before blood was withdrawn?

Results: Text describes in detail the obtained values and their significance in drawing conclusions. Data are presented in an appropriate way.

§  Line 109: “Patients were 60±1 years old.” In this context, it would be beneficial to utilize the Interquartile Range (IQR). The IQR is less sensitive to extreme outliers in the data compared to measures like standard deviation.

§  Were there any limitations or potential confounding factors that should be considered when interpreting these results?

Discussion: Relevance and importance of the obtained results are explained well in the discussion section. The results are discussed from multiple angles and placed into context without being overinterpreted. However:

§  Can the authors elaborate on the potential reasons behind the lack of significant correlations between serum KL-6 levels and demographics or routine laboratory markers in their cohort? Are there any subgroup analyses or additional factors that could explain these findings?

§  Regarding the discussion of serum KL-6 levels on day 3 and its potential predictive value, could the authors explain in more detail the mechanisms behind this observation? What is the clinical significance of KL-6 reflecting regenerative processes on day 3?

§  The focus of your research revolves around dynamic changes. In the context of contemporary diagnostics, it is advisable to consider incorporating a discussion on the potential utility of AI algorithms, such as artificial neural networks, for tracking dynamic changes e.g. PMID: 36972470, 37238309, 37515208. This could enrich your research by providing valuable insights into future perspectives. It not only aligns with contemporary diagnostic trends but also has the potential to significantly improve patient care, research methodologies, and the overall quality of healthcare delivery.

§  Given the complexity of KL-6 dynamics and its potential clinical relevance, are there any recommendations or future research directions the authors suggest for further investigating the role of KL-6 in SARS-CoV-2 pneumonia or other pulmonary diseases?

Conclusion: Conclusions answer the aims of the study and are fully supported by the results.

References: References come from reputable sources and all are cited properly throughout the manuscript.

This article tackles an important subject and presents valuable results. While there are areas that could be strengthened, the article lays a solid foundation for future research in the field. With the suggested improvements, this article has the potential to contribute significantly to our understanding of the role of the KL-6 in hospitalized patients with SARS-CoV-2 pneumonia.

Author Response

Comments of Reviewer 1

  1. Reviewer´s suggestion:

Introduction: The introduction provides a clear and concise overview of the research problem, establishing its relevance in the field. The opening paragraph effectively captures the reader's attention by highlighting the significance of the topic. However:

  • Lines 33-34: “An easy-to-obtain biomarker to stratify patients at risk of poor outcome during SARS-CoV-2 pneumonia remains an unmet need.” Which laboratory parameters have been utilized up to this point, particularly those that are part of routine clinical laboratory diagnostics? E.g. Refer to PMID: 35976368 and 36371787. Highlighting the importance of this information is crucial because understanding which laboratory parameters have been used and why they are relevant provides valuable context for the study and helps establish the basis for comparing the effectiveness of KL-6 in clinical assessment. This knowledge aids in assessing the significance and potential clinical impact of the research findings.

Authors´ statement:

We thank the Reviewer for this comment. We have added the requested information in the introduction:

Several studies have reported patients characteristics such as body-mass index, immune-inflammatory parameters, such as procalcitonin (PCT), c-reactive protein (CRP), lactate dehydrogenase (LDH), interleukin-6 (IL-6) and peripheral blood cell counts, such as leucocytes, lymphocytes, neutrophil granulocytes, thrombocytes or the ratio between different laboratory parameters, such as neutrophil-lymphocyte ratio (NLR) as potential predictors for severe SARS-CoV-2 disease [2, 4-6]. All of this parameter are not disease specific or lung specific and often have better prognostic value when considered together.

  1. Reviewer´s suggestion:

Methods: The study methods are valid and reliable. The process of subject selection is clear. Variables are defined and measured appropriately. The authors employed appropriate statistical methods for the treatment of the data obtained. However:

  • Did author employ the power analysis?

Authors´ statement:

We thank the Reviewer for this question. We did not employ the power analysis. We added this point to the limitations of the study. We included all consecutive patients who were hospitalized between April 2020 and August 2021 and gave their written informed consent. This implies that dying patients or patients who could not give consent for other medical reasons were excluded. We added a table (table S1) with our inclusion and exclusion criteria to Supplemental Materials.

  1. Reviewer´s suggestion:
  • Which method was used to determine whether sample data has been drawn from a normally distributed population?

Authors´ statement:

We thank the Reviewer for the appropriate comment. We have checked the distribution of all variables before performing statistical analysis by using Kolmogorov-Smirnoff test. “Non normal distribution was observed for all variables but body-mass index (BMI), LDH, CRP and neutrophil count in Kolmogorov-Smirnoff test.” We have added this information in the text of the results.

  1. Reviewer´s suggestion:
  • Please provide explicit specifications for both the criteria that should be considered for inclusion and those that should be excluded.

Authors´ statement:

We thank the Reviewer for this request. We included all Patients who were hospitalized between April 2020 and August 2021, gave their written informed consent for data research and had a SARS-CoV-2 pneumonia. We did not exclude anyone a priori, in order to achieve a broad cross-section of those with COVID. We added a table with our inclusion and exclusion criteria in Supplemental Material (table S1).

  1. Reviewer´s suggestion:
  • Lines 81-83: “Other routine laboratory parameters (leucocytes, lymphocytes, neutrophil granulocytes, thrombocytes, procalcitonin (PCT), c-reactive protein (CRP), lactate dehydrogenase (LDH), interleukin-6 (IL-6) were measured in fresh heparinized blood or plasma.” Please provide information on the methodology and the specific analyzer used to determine these parameters.

Authors´ statement:

We thank the Reviewer for this thoughtful comment. We amended the text as following:

“Other routine laboratory parameters (leucocytes, lymphocytes, neutrophil granulocytes, thrombocytes, procalcitonin (PCT), c-reactive protein (CRP), lactate dehydrogenase (LDH), interleukin-6 (IL-6) were measured in fresh heparinized blood or plasma. Absolute numbers of leukocytes, lymphocytes, neutrophil granulocytes, and thrombocytes (platelets) were measured by using fluorescence flow cytometry in Sysmex XP300 Automated Hematology Analyzer (Sysmex, Norderstedt, Germany). LDH, PCT, and CRP was analyzed by Atellica Solution Immunoassay & Clinical Chemistry Analyzers (Siemens Healthineers International AG, Zurich, Switzerland). IL-6 was determined with a sequential solid phase chemiluminescence immunoassay by IMMULITE 2000 XPI Immunoassay Analyzer (Siemens Healthineers International AG, Zurich, Switzerland).”

  1. Reviewer´s suggestion:
  • Line 78: “Blood was taken on the day of admission.” Was there any specific treatment given to the patients upon admission before blood was withdrawn?

Authors´ statement:

We thank the Reviewer for pointing this out. “Blood was withdrawn within 24h after the triage, mostly before any COVID-19-specific treatment was administered. We cannot exclude that patients have taken other medications at home, such as anti-inflammatory or pain relievers.” We have clarified this point in the methods.

  1. Reviewer´s suggestion:

Results: Text describes in detail the obtained values and their significance in drawing conclusions. Data are presented in an appropriate way.

  • Line 109: “Patients were 60±1 years old.” In this context, it would be beneficial to utilize the Interquartile Range (IQR). The IQR is less sensitive to extreme outliers in the data compared to measures like standard deviation.

Authors´ statement:

We thank the Reviewer for this suggestion, we have amended the text accordingly.

  1. Reviewer´s suggestion:
  • Were there any limitations or potential confounding factors that should be considered when interpreting these results?.

Authors´ statement:

We thank the Reviewer for this comment. Compared to the previous version of the manuscript, we have make a clearer statement about the limitations of the study. The study has several limitations. “First, the small sample size, which did not allow performing subgroup analyses. Second, we did not employ power analysis, thus we do not know if statistical power is sufficient. Third, we did not systematically review high resolution CT scans, so that we were not able to adjust the predictors for pneumonia extent. Fourth, this was a retrospective exploratory analysis. Finally, the serial measurements of KL-6 during hospitalization were performed only in few patients, limiting the value of our longitudinal analysis.”

  1. Reviewer´s suggestion:

Discussion: Relevance and importance of the obtained results are explained well in the discussion section. The results are discussed from multiple angles and placed into context without being overinterpreted. However:

  • Can the authors elaborate on the potential reasons behind the lack of significant correlations between serum KL-6 levels and demographics or routine laboratory markers in their cohort? Are there any subgroup analyses or additional factors that could explain these findings?

Authors´ statement:

We thank the Reviewer for these relevant questions.

“The lack of significant correlations between KL-6 levels and demographics has been already pointed out in other studies [25-27]. This can be seen as an advantage, that KL-6 levels do not depend on age, gender BMI or smoking status, reducing the possible influence of covariates on the results of the present study.” We added this point in the discussion.

With regard to the second concern, our study was underpowered to perform subgroup analyses, such as those to unravel the influence of additional factors on the observed correlations. To overcome this problem we have added as many covariates as possible to the multistep logistic regression analysis. We added this point to the limitations.

  1. Reviewer´s suggestion:
  • Regarding the discussion of serum KL-6 levels on day 3 and its potential predictive value, could the authors explain in more detail the mechanisms behind this observation? What is the clinical significance of KL-6 reflecting regenerative processes on day 3?

Authors´ statement:

We thank the Reviewer for this helpful comment. We have added these:

„KL-6 is essentially produced by regenerating pneumocyte type II and gets into the blood by increased permeability following disintegration of the alveolar-vessel barrier after injury [7]. The kinetic of elevation of KL-6 in serum over time has not been sufficient explored, but we hypothesize a similar pathway as for acute exacerbation of idiopathic pulmonary fibrosis [9]. We cannot exclude that treatment initiated between day 1 and 3 could have significantly affected the elevation of KL-6 levels in serum. Due to the low number of serial KL-6 measurements, we cannot provide an unequivocal interpretation of these findings. “

  1. Reviewer´s suggestion:
  • The focus of your research revolves around dynamic changes. In the context of contemporary diagnostics, it is advisable to consider incorporating a discussion on the potential utility of AI algorithms, such as artificial neural networks, for tracking dynamic changes e.g. PMID: 36972470, 37238309, 37515208. This could enrich your research by providing valuable insights into future perspectives. It not only aligns with contemporary diagnostic trends but also has the potential to significantly improve patient care, research methodologies, and the overall quality of healthcare delivery.

Authors´ statement:

We thank the Reviewer for this interesting suggestion and we agree that this should be the subject of future investigations. We added this point in the discussion as further perspective to implement biomarker studies.

“A further perspective to implement biomarker studies are AI algorithms, such as artificial neuronal networks, for tracking dynamic changes. We learned from several studies that automated machine learning algorithms improve finding clinical diagnosis by been objective, more efficient. This can facilitate the daily clinical routine and can contribute to find reliable biomarkers. AI algorithms have the potential to significantly improve research methodologies and as a result patient care and quality of healthcare delivery. Consideration should be given in future biomarker studies.”

  1. Reviewer´s suggestion:
  • Given the complexity of KL-6 dynamics and its potential clinical relevance, are there any recommendations or future research directions the authors suggest for further investigating the role of KL-6 in SARS-CoV-2 pneumonia or other pulmonary diseases?.

Authors´ statement:

We added this important aspect to the implementation strategies for biomarkers studies - as per comment 11- in the discussion. 

Reviewer 2 Report

Comments and Suggestions for Authors

The authors monitored the serum biomarker and compared moderate to severe COVID-19, including pneumonia. The study is interesting. Here is my comment to improve the manuscript.

1. Some typing errors as follows,

- Table 1, 2, 5, correct ‘IV (yes, no)’ to ‘IV (yes/ no)’.

- Correction of the p-value of PCT (in Line 122) and Table 2.

- Line 168 changes “ KL-6 serum levels” to “Serum KL-6 levels”

2. Figure 3 requires additional explanation. Many colored lines need to be labeled. It seems like some of the serum was selected to show in this figure.  There should be an address somewhere on how to choose the serum.

3. Again, regarding Table 2, a subgroup of serum samples was selected to monitor KL-6 and other routine testing. The varying number of tested sera, such as KL6 at the based line, day 3, and day 7, could introduce bias selection into the results.

4. Please check and verify Table 2; I have found the two P-values in the marker LDH and CRP.

5. Please review the total number of serum LDH in Table 3, as it does not match Table 2 (n=81+58 = 139, not 138). Additionally, you may add an asterisk (*) after 0.004* to help the reader see easily.

6. You may re-word the sentence (lines 185-186) “KL-6 levels at baseline positively correlated with LDH serum levels (r=.244, p=.004) only (table 3)”  to “ The baseline KL-6 levels exhibited a positive correlation with LDH levels (r=.244, p=.004), while IL-6, PCT, CRP, NLR, and P/F levels did not show any significant correlation (p>0.05) (Table 3).”

7. Line 267-268: You should carefully recheck the wording: “We did not find any significant correlation between serum KL-6 levels and demographics or with other routine laboratory markers (table 3) in our cohort.” It seems incorrect. You say that ‘baseline KL-6 correlated to LDH in result section’. You may provide a narrative discussion on this marker.    

Comments on the Quality of English Language

-

Reviewer 3 Report

Comments and Suggestions for Authors

The authors have reported in the present study that serum KL-6 levels at hospital admission are higher in patients with SARS-CoV-2 pneumonia and severe outcome. Additionally, they identified a cut-off of serum KL-6 with the potential to predict severe SARS-CoV-2 pneumonia outcome compared to other routine biomarkers. They have used good statistical analysis and have also indicated that CRP, PCT, LDH and NLR, in line with other investigations were good predictors for SARS-CoV-2 pneumonia outcome. The study is very extensive and the results are also very promising. The outcome of the study also indicated that serum KL-6 levels on day 3 after admission was equally high in both moderate and severe groups. On day three, it is impossible to decide which group the patient belongs to. Thus, the KL-6 level must be monitored on admission and treatment strategy must start according to the WHO scores for moderate and severe disease. I have a few comments:

Technical comments:

In Table 1, 2, and 5. what does the parameter “Delta KL-6, U/ml baseline to day 7 “ mean? Please explain in the footnotes. In the text it is mentioned that” The difference in KL-6 levels between day 0 and day 7 is Delta KL-6”, but in the table if we subtract the average value of KL-6 at day 0 from day 7 value, the results are different that what is stated in the tables (1, 2, 5). Please explain how did the authors derive this value in the methodology section.

Other comments:

In table 2, Gender should be M/F not m/f. it should be corrected.

The English language needs to be checked thoroughly, especially in the section 2. Statistical Analysis.

There are many small paragraphs in the discussion section and I suggest them to combine together to make a single paragraph.

I recommend minor revision.

Comments on the Quality of English Language

The English language needs to be checked thoroughly, especially in the section 2. Statistical Analysis.

Round 2

Reviewer 1 Report

Comments and Suggestions for Authors

The authors omitted all of the recommended references.